# Nanotopography and Microconfinement Impact on Primary Hippocampal Astrocyte Morphology, Cytoskeleton and Spontaneous Calcium Wave Signalling

**DOI:** 10.3390/cells12020293

**Published:** 2023-01-12

**Authors:** Anita Previdi, Francesca Borghi, Filippo Profumo, Carsten Schulte, Claudio Piazzoni, Jacopo Lamanna, Gabriella Racchetti, Antonio Malgaroli, Paolo Milani

**Affiliations:** 1CIMaINa and Dipartimento di Fisica, Università degli Studi di Milano, Via Celoria 16, 20133 Milano, Italy; 2Centre for Behavioural Neuroscience and Communication (BNC), Università Vita-Salute San Raffaele, 20132 Milano, Italy; 3Division of Neuroscience, Scientific Institute Ospedale San Raffaele, 20132 Milano, Italy

**Keywords:** astrocytes, nanostructured zirconia, micrometric pattern, phenotypes, calcium waves

## Abstract

Astrocytes’ organisation affects the functioning and the fine morphology of the brain, both in physiological and pathological contexts. Although many aspects of their role have been characterised, their complex functions remain, to a certain extent, unclear with respect to their contribution to brain cell communication. Here, we studied the effects of nanotopography and microconfinement on primary hippocampal rat astrocytes. For this purpose, we fabricated nanostructured zirconia surfaces as homogenous substrates and as micrometric patterns, the latter produced by a combination of an additive nanofabrication and micropatterning technique. These engineered substrates reproduce both nanotopographical features and microscale geometries that astrocytes encounter in their natural environment, such as basement membrane topography, as well as blood vessels and axonal fibre topology. The impact of restrictive adhesion manifests in the modulation of several cellular properties of single cells (morphological and actin cytoskeletal changes) and the network organisation and functioning. Calcium wave signalling was observed only in astrocytes grown in confined geometries, with an activity enhancement in cells forming elongated agglomerates with dimensions typical of blood vessels or axon fibres. Our results suggest that calcium oscillation and wave propagation are closely related to astrocytic morphology and actin cytoskeleton organisation.

## 1. Introduction

Mammalian brains are constituted, for a consistent fraction, by glial cells (microglia, astrocytes and oligodendrocyte) [1]. In the past, glial cells were believed to fulfil an ancillary role, just providing mechanical and metabolic support to neurons. More recently, it became clear that glial cells can communicate very effectively thus affecting directly and indirectly the functioning and the fine morphology of the brain, both in physiological and pathological contexts [2,3]. In particular, astrocytes play a fundamental role in the homeostatic control of the concentration of ions, bioactive molecules and water in the extracellular space, and in the modulation of neurotransmission (tripartite synapse) as structural components for synaptic plasticity, formation of memory, learning, regulation of sleep or metabolisms [4,5].

Astrocytes are fundamental for the correct functioning of neurons, as demonstrated by many pathological conditions where their involvement has been reported, including neurodegenerative and demyelinating diseases, epilepsy, trauma, ischemia, infection and cancer [6]. Although the triggering causes of these brain diseases are multifactorial (e.g., genetic, trauma, infection, environmental health, tissue mechanics, etc.), tissue morphology and mechanics are reported as the major driving forces in the development of these pathological state [7,8,9]. 

The morphology and the positioning of astrocytes is related to their ability to perceive both mechanical and morphological cues of the microenvironment through mechanotransductive processes [10,11,12]. They sense certain extracellular cues and adapt to the physical properties of the extracellular matrix (ECM), based on its specific stiffness and composition [7,13,14,15,16,17,18,19,20]. From a functional point of view, astrocytes form intercellular networks in which information is propagated with signalling mechanisms on temporal and spatial scales different from the neuronal counterpart [21]. At the same time, they communicate with neurons, integrate information processing [22,23] and have an active role in modulating neuronal synaptic connectivity [24]. Astrocytic communication is based on the integration of intracellular and extracellular signalling pathways, meaning that when activated internally they can signal to the environment, especially via ion exchanges and neurotransmitter release. Their activation, usually detected as a transient or oscillatory increase in cytosolic Ca^2+^, can be propagated to neighbouring cells via cell to cell channels, i.e., the gap junctions, which mediate the propagation of waves in a network of glial cells [25]. Such signalling events can apparently occur spontaneously or can be triggered by external stimuli. The properties of the Ca^2+^ signalling cascade have been shown to be strongly dependent both on morphological parameters of the single astrocyte [26,27] and on the spatial organisation of their networks [2,28,29,30].

Although many aspects of the role of astrocytes are well understood and characterised, their functions in the brain under both physiological and pathological conditions remain, at least to a certain extent, elusive, especially regarding their participation in brain cell signalling. In vitro studies of primary astrocytes, although not recapitulating the complex physiological 3D brain environment, can be of help if some aspects of the morphological and mechanical properties of the ECM can be reproduced at the nano- and microscale mimicking the native environment that astrocytes find in the brain, e.g., in terms of nanotopographical surface properties and geometrical features [31].

Recently, we demonstrated the fabrication of complex micropatterns of nanostructured zirconia films able to control cell adhesion and their spatial distribution [32,33]. Zirconia nanostructured substrates have been used in a variety of experimental studies [32,34,35,36,37,38,39] to provide cellular microenvironments with controllable and reproducible biomimetic nanotopographies to unravel their impact on the mechanisms underlying mechanotransduction and to guide and control cellular behaviour [32,33]. Furthermore, these nanotopographical surfaces have been tested and validated as micropatterned substrates on a neuronal-like cell line (PC12), CA3-CA1 primary hippocampal neurons and astrocytes. For the neuronal cells, it has been shown that they grow and differentiate in a confined manner in these nanostructured zirconia micropatterns [32,33]. For astrocytes, so far the effective confinement of astrocytes in a nanostructured zirconia pattern has also been demonstrated in Ref. [33]. An assessment of the influence of nanotopography and micropatterns on astrocyte behaviour and functioning has not yet been investigated. 

Here, we report the effect that these nanostructured zirconia micrometric patterns and the consequential confinement of primary astrocytes has on astrocyte morphology, organisation and calcium signalling. The impact of astrocyte/nanotopography interaction and restrictive adhesion, in terms of geometry, on several morphological properties of single astrocytes, the actin cytoskeleton, the network organisation and calcium-related activity, was identified. In addition, an analysis of the propagation of calcium waves reveals a possible influence of the geometric organisation of cells on calcium wave events.

## 2. Materials and Methods

### 2.1. Fabrication of Micropatterned Nanostructured Substrates

The fabrication of micrometric patterns for the confined culture of astrocytes involved the functionalisation of glass coverslips with an antifouling molecule, PAcrAm-g-(PMOXA, NH2, Si, produced by SuSoS AG 151—Dubendorf, Switzerland), and the subsequent fabrication of a nanostructured zirconia (ZrO_x_) coating via supersonic cluster beam deposition (SCBD), according to a specific micrometric pattern. The detailed protocol for the substrate cleaning and antifouling functionalisation is described in [33]; here, we briefly describe the experimental details of nanostructured pattern deposition.

Cluster-assembled zirconia micropatterns were deposited on passivated coverslips using an SCBD apparatus equipped with a pulsed microplasma cluster source (PMCS) [40]. A zirconium rod placed in the PMCS was ablated via a pulsed plasma discharge ignited by the injection of Ar pulses; the ablated species were thermalised with the injected inert gas and condensed to form zirconia clusters and the clusters–Ar mixture was then expanded through a nozzle into a vacuum to form a supersonic beam [41]. The cluster beam impinges on a substrate placed in a second vacuum chamber. Deposition process parameters were controlled in order to select specific characteristics of the resulting film and, in particular, the nanoscale surface roughness [42,43], defined as the standard deviation of the heights. This approach produced nanostructured films grown by a ballistic deposition [42,44], which resulted into a nanoscale topography whose roughness (Rq) could be accurately controlled and varied in a reproducible manner according to the thickness (t) of the thin film, using a simple scaling law Rq~t^β^ [42,43]. We produced continuous nanostructured zirconia films with 10 and 15 nm surface roughness, and micropatterns composed of triplets of ZrO_x_ dots of 100 μm in diameter with an interaxial distance of 500 μm and connected by bridges 50 μm and 20 μm wide. We also fabricated smooth zirconia films (surface roughness < 1 nm), deposited via means of an electron beam evaporator to be used as a reference sample (flat ZrO_x_). 

The micrometric features characterising the design of the pattern were obtained by the use of a stencil mask produced by photoresist lithography (PRL) of thin silicon wafers [33]. 

### 2.2. Culture of Primary Astrocytes

We cultured primary astrocytes derived from the hippocampus of neonatal Sprague Dawley rats (Charles River Laboratories Italia). All the procedures were performed according to the research and animal care procedures approved by the institutional animal care and use committee for good animal experimentation of the Scientific Institute San Raffaele complying with the code of practice for the care and use of animals for scientific purposes of the Italian Ministry of Health—IACUC Number 728. P2-P5 rats of both sexes were rapidly decapitated, the hippocampi from both hemispheres were dissected out and cells were mechanically and chemically dissociated as previously described [45]. After extraction, the cells were maintained in MEM medium supplemented with 10% foetal calf serum, 33 mM of glucose, 2 mM of Glutamax and 2 U/mL of Penicillin-Streptomycin (all reagents were obtained from Thermo Fisher Scientific, Gibco Massachusetts, USA, if not stated otherwise). Cells were grown on standard culture substrates for 12 DIV at 37 °C and 5% CO_2_, and every 3 days the culture medium was replaced. Cells were then detached with a trypsin/EDTA solution and, after centrifugation (1000 rpm, 4 °C for 5 min), the pellet was resuspended. The cells were subsequently seeded on the patterned substrates with a density of 12,500 cells/cm^2^. On these substrates, cells were kept in the same culture medium described above for 1 day, replacing the medium the day after. On Day 2, the medium was replaced with one with a 1% concentration of foetal calf serum.

### 2.3. Immunofluorescence Imaging

All the reagents were purchased from Merck KGaA, Darmstadt, Germany, if not stated otherwise. For the immunofluorescence imaging, we used astrocytes grown on nanostructured films and patterned substrates for 3 days. We fixed them with 4% paraformaldehyde (PFA)/phosphate buffered saline (PBS) for 10 min. We then permeabilised the cell membranes with 0.2% Triton X-100/PBS for 3 min and blocked them with 3% bovine serum albumin (BSA)/PBS. Phalloidin and tetramethylrhodamine (TRITC) conjugate were used to stain the actin cytoskeleton of the cells and Hoechst 33342 was used for the nucleus. They were incubated for 45 min in a humid environment at room temperature. After the staining, the cells were mounted with ProLong^®^ Gold antifade (MolecularProbes, Eugene, OR, USA). Images were taken using a confocal (Nikon A1R, Tokyo, Japan) or combined confocal and 3D Structural Illumination Microscope (Nikon A1) with objectives 2×, 4×, 10× and 100× at the UNI^TECH^ NOLIMITS Imaging Facility of the University of Milano.

### 2.4. Characterisation of the Morphological Properties of the Astrocytes

We cultured astrocytes on zirconia flat and nanostructured substrates for 5 DIV, and we took phase contrast images on Day 1, 3 and 5. Initial seeding density was the same for every sample. Cell density was calculated by manually selecting the cells. The analysis was carried out on at least 10 images for each condition. Initial seeding density was low enough to guarantee that the culture would not be confluent on Day 5, in order to be able to analyse the shapes of independent cells.

The phase contrast images were taken with a microscope (Axiovert 40 CFL—Zeiss, Oberkochen, Germany) equipped with 20×/0.3 ph1, CP-ACHROMAT 10×/0.25 Ph1 and 5×/0.12 CP-ACHROMAT objective and with a high-definition photo camera (TiEsseLab TrueChrome HD II, Milan, Italy) operated using ISCapture imaging software (version 3.6.9.3).

The cell contour was manually extracted using the Image Processing Toolbox from MATLAB (2020b, The MathWorks Inc., Natick, MA, USA), and in particular the Image Segmenter App. Then, with custom-made scripts developed using MATLAB, we calculated the cell area, cell complexity and cell shape index (CSI) parameter. The cell area was calculated as the number of pixels inside the cell contour, and was then converted to μm^2^. We first skeletonised the images of a filled cell contour; this means we extracted the centreline while preserving the topology of the object. To do this, we used the built-in MATLAB function *bwskel*, that reduces all objects in a 2D binary image to 1-pixel-wide curved lines, without changing the essential structure of the image. The complexity was the number of endpoints of the cell skeleton, which corresponded to the number of cell protrusions. The CSI parameter was defined as $CSI=\frac{p^{2}}{4\pi A}$, where p is the cell perimeter, i.e., the number of pixels in the cell contour, and A is the cell area, i.e., the number of pixels inside the cell contour. The CSI parameter was =1 for perfectly round cells, and >1 for elongated or ramified cells.

To extract quantitative parameters describing the actin cytoskeleton, we converted the immunofluorescence 3D stacks to a single 2D image by performing a maximum z-projection, using imageJ (NIH, New York, NY, USA). This function was attributed to every pixel, with the maximum intensity of the pixels being over all of the images in the stack. 

To calculate actin coverage, we defined a unique greyscale intensity interval to which every image was mapped, to allow comparison between images. The range boundaries were taken as the highest and lowest greyscale intensities among the range boundaries of all of the images analysed. We then set a threshold to 25% of the maximum intensity registered, and we binarised the images by setting the pixels whose intensity was higher than the threshold to 1. The number of ones per cell in every image was defined as the actin intensity. We used the Image Segmenter tool (MATLAB) to manually extract the cell borders and to calculate the pixel area of every cell. The actin coverage was calculated as the percentage of the actin intensity over the cell area. 

To extract the width of actin fibres, we extracted 1D intensity profiles of the actin cytoskeleton and a profile of the background intensity. A baseline value was defined as the mean value of the background profile. From the actin profiles, actin fibres were identified as peaks with prominence higher than 3 times the baseline value. In particular, the actin fibre widths corresponded to the width at half prominence of the peak (extracted using the MATLAB function findpeaks). We extracted 3 profiles for every cell image analysed. For the quantification of both the actin coverage and the actin fibre widths, we analysed 10 cells for every condition. We repeated each experiment three times, and for every experiment we had at least 3 replicates for each type of substrate.

### 2.5. Calcium Imaging and Analysis of Calcium Wave Propagation

Hippocampal astrocytes grown for 5 days on nanostructured films and micropatterned substrates were loaded with 2 μL of 2.5-μM-concentrated Fluo-4AM loading solution (Thermo Fisher Scientific, Invitrogen, Waltham, MA, USA) for 30 min at 37 °C, 5% CO_2_. During the experiment, the cells were kept at room temperature, submerged in a Tyrode solution containing 119 mM NaCl, 5 mM KCl, 2 mM CaCl_2_, 2 mM MgCl_2_, 25 mM HEPES (4-(2-hydroxyethyl)-1-piperazineethanesulfonic acid) and 30 mM D-glucose (pH adjusted to 7.4 with NaOH, osmolarity adjusted to 300 mOsm). O_2_ was bubbled into the solution for the entire duration of the experiment. The sample was placed in a customised chamber with tubes enabling constant replacement of the Tyrode solution. We recorded several 5 min time-lapse fluorescence videos using an inverted epifluorescence microscope (Axiovert 135, Zeiss, Oberkochen, Germany), equipped with a standard mercury arc lamp, filter set for fluorescein isothiocyanate (FITC) and a digital camera (Orca-ER, Hamamatsu, Shizuoka, Japan). Images were taken every 0.4 s, a typical time scale for characterising slow calcium signalling activity. The characterisation of faster calcium transients, obtained using a more sophisticated apparatus, is planned for the future.

We analysed the calcium imaging videos using custom-made scripts developed in MATLAB (2020b, The MathWorks Inc., Natick, MA, USA). The fluorescence traces were extracted as the sum of the intensity of pixels within circular regions of interest (ROI) of equal radius, with the centre corresponding to the most responsive cell soma, manually segmented. Each trace was normalised by the baseline intensity value, and the exponential background was subtracted. We also performed a 10-point box smoothing of the traces to reduce the high-frequency noise. 

The fluorescence traces displayed peaks and modulation in correspondence to intracellular calcium elevations. In particular, we quantified the spontaneous calcium activity of astrocytes grown on continuous (not patterned) flat and 15-nm-rough zirconia substrates. 

We also characterised the correlated events (propagation between neighbouring cells) of spontaneous calcium signals taking place between cells belonging to the same connected micrometric pattern by considering the modulations in calcium traces only if their fluorescence intensity was higher than a certain threshold, defined as the standard deviation of the baseline noise multiplied by 10. We attributed to every peak detected a time *t_I_* at which the intracellular calcium elevation begins, defined as the instant of time at which the fluorescent trace has an intensity equal to half of the maximum peak intensity. In fact, peaks typically display a smoothly rounded shape at the time corresponding to the maximum intensity values; thus, it was difficult to extract a unique time instant for the peak maximum. The initial calcium elevation, however, is fast with respect to the video sampling rate, so a unique time instant corresponding to the half-height could be extracted more easily. Here, we defined a calcium wave event as a set of peaks, identified on different cell traces, which have space–time properties that are compatible with calcium propagation. 

Experimentally observed calcium waves in cultured astrocytes and slices have a velocity of propagation between 15–27 μm/s and a maximal propagation range of 200–350 μm [46,47,48,49]. Therefore, two cells were considered to participate in a calcium wave event if their mutual distance was smaller than 350 μm and calcium events appeared within a time interval of 23 s, which corresponds to the time needed to travel 350 μm at the minimum velocity of 15 μm/s. Furthermore, calcium wave events that travelled at a velocity >27 μm/s were excluded. We made the assumption that the calcium wave could only travel radially [50] away from the source cell, i.e., the first cell exhibiting a peak. The subsequent responding cells were taken as secondary sources which could propagate signals radially away from the primary source. Since calcium elevations can be propagated extracellularly via the diffusion of ATP [51,52], we also assumed that the non-nearest neighbours could exchange signals. This analysis was performed using custom scripts developed in Igor Pro (Wavemetrics, Lake Oswego, OR, USA).

### 2.6. Statistical Analysis

Statistical comparisons were performed using the Kruskall–Wallis test [53] followed by multiple post hoc comparison analyses, carried out using Bonferroni’s method [54]. The difference was considered statistically significant if the *p* value was <0.05, indicated with the symbol *. *p* value < 0.01 was indicated as **, and *p* value < 0.001 was indicated as ***.

## 3. Results and Discussion

### 3.1. Micrometric Patterns with Nanostructured Zirconia Films

We used homogeneous and micropatterned nanostructured zirconia thin films as substrates to study the influence of nanotopography and microconfinement on primary hippocampal rat astrocytes.

Figure 1a shows an image acquired using the optical microscope of the two different motifs of the micrometric patterns with nanotopographical substrates printed on the coverslip. The precise control of the nanoscale topography can be easily obtained over macroscopic areas, as required for the large number of experiments typical of in vitro biological assays [35,36], with a precise resolution of the micrometric pattern. In particular, the morphology of the homogenous deposit is composed via a granular and porous matrix of cluster-assembled material (Figure 1b), whose surface roughness is due to the cluster dimension and to the growth of the film in the ballistic deposition mode [43]. We produced micropatterns with island and bridge elements, the latter with different widths (20 and 50 µm). The gradient of height along the bridges which connects islands extends approximately 10 µm from the plateau characterising the homogeneous film (Figure 1c,d).

Astrocytes and their processes often surround CNS blood vessels, synaptic areas [55] or axon tracts [56]. These brain features have similar shapes (i.e., elongated structures) to the bridges in our patterns. The dimensions are also comparable to the diameters of the blood vessels in the human brain ranges, e.g., between 8 μm and 500 μm [57,58], and the typical diameters of axon fibre ranges between 0.16 μm and 9 μm [59]. At the same time, the nanostructured surface morphology of the micropatterns may be particularly suitable to mimic the in vivo environment for astrocytes as, from an ultrastructural point of view, the nanotopography produced by SCBD is similar to the one found in the ECM surrounding blood vessels, i.e., a basement membrane [60]. 

### 3.2. Astrocyte/Nanotopography Interaction Induces Morphological, Cytoskeletal and Functional Differences

First, we assessed the effect of astrocyte interaction with the nanotopographical features (using homogeneous substrates with different roughness parameters) on different cellular features (morphology and cytoskeletal organisation) and functioning (calcium activity).

In vivo astrocytes are star-shaped glial cells. However, this morphology observed in vivo is not always found in in vitro cell cultures. In fact, astrocytes cultured on two-dimensional flat glass or tissue culture plastic often display a spread morphology that closely resembles the morphology of reactive astrocytes found boarding the brain lesion [61]. 

A morphological analysis of astrocytes that were seeded on a variety of topographies was performed to investigate whether the nanotopographical substrate features have an effect on astrocyte morphology and cell shape complexity. 

The astrocyte morphology was evaluated by analysing different morphological parameters, such as the cell area, the complexity (calculated as the number of distal processes) and the CSI (cell shape index), a parameter that is equal to 1 for perfectly round cells and is larger than 1 for ramified and elongated morphologies [62] (see Materials and Methods for in-depth description). The optical images acquired of astrocytes grown on different substrates for 5 days and the corresponding morphological properties on Day 1, Day 3 and Day 5 are illustrated in Figure 2. 

For Day 1, no differences are statistically significant; on Day 3, there is an intermediate situation with only some parameters being significantly different, whereas after 5 days all of the analysed parameters display significant differences. Astrocytes cultured on nanostructured ZrO_x_ substrates possess smaller cell areas and adopt more ramified cell shapes, as confirmed by the increased complexity and CSI parameter. The distributions of the values of complexity and CSI are also wider on the substrates of roughness 15 nm, testifying greater cell morphology variability.

The ramified and more complex cell morphology seen on cluster-assembled films resembles more closely the astrocyte appearance in vivo [61]. Our findings are congruent with previous studies which demonstrated the promotion of more physiologically relevant phenotypes in astrocytes via substrates assembled with nanoscale building blocks, such as nanofibres [63], nanofibre-based 3D culture systems [64] or water-repellent fractal tripalmitin (PPP) surfaces [65]. Morphologies similar to those observed in nanostructured substrates were also recapitulated by astrocytes cultured on gels, with a stiffness of the same order as brain tissue [66,67], always related to the mechanotrasductive mechanism. Therefore, our results might indicate a role of the astrocyte/nanotopography interaction related to cell adhesion and mechanotransductive events (as has been shown for neuronal cell types [35,37,39]).

To get an idea as to whether the astrocytic interaction with nanotopographical features affects an essential component and mediator of the mechanotransductive machinery, we characterised the actin cytoskeletal organisation with dependency on different roughness parameters on Day 3. The time point was chosen to understand whether actin cytoskeletal remodelling might precede the significant manifestations in astrocyte morphology (which were present on Day 5). We performed a staining of the F-actin of fixed astrocytes and recorded images using a 3D Structural Illumination Microscope (3D-SIM). We quantified the differences in actin organisation of astrocytes on the different substrates by measuring the percentage of cell area with a fluorescence signal coverage higher than 25% of the maximum coverage measured over all of the samples. Moreover, we extracted the width of the stress fibres.

The fluorescence images reported in Figure 3a show that cells display a change in cytoskeletal organisation on the nano-rough surfaces (images in the second column correspond to 10 nm surfaces, those in the third column correspond to 15 nm surfaces), compared to the flat substrate condition (images in the first column).

Astrocytes cultured on flat surfaces display an intense fluorescence signal of reticulated F-actin covering a big fraction of the cell with distinct stress fibres. In contrast, the signal is much less intense on the nanotopographical surfaces, and we also detect fewer and thinner stress fibres. The results are reported in the boxplot in Figure 3b,c. In Figure 3b, we observe that on flat surfaces, the actin cytoskeleton coverage has a median value of 46%, as opposed to 14% and 12% on substrates of roughness of 10 nm and 15 nm, respectively. Astrocytes also display thicker actin fibres on flat substrates compared to the rough ones, as reported in Figure 3c. 

These data accentuate decisive differences in the architecture of the actin filaments of the cytoskeleton with dependency on the astrocyte/nanotopography interaction. Interestingly, these changes in the actin cytoskeleton can be seen before (on Day 3) the morphological modulations became evident (on Day 5). 

Schulte et al. found similar results with neuron-like PC12 cells cultivated on nanostructured ZrO_x_ substrates [37]; they demonstrated that the formation of high-order actin filament structures (e.g., stress fibres) were mainly established on flat ZrO_x_. A similar trend was shown in human pancreatic β-cells, whose stress fibres were mainly detected on flat ZrO_x_, but seldom formed on nanostructured ZrO_x_, where actin was organised in bundles on the cell–cell contacts [68]. This nanostructure-induced effect was found to be related to the geometry of the cell/substrate contacts which affects essential mechanotransductive processes and structures, such as the dimensions of integrin-mediated adhesion sites [37,68], force loading in molecular clutches [39,69] and the composition of the actin cytoskeleton, which in turn dictates the biomechanical and morphological properties of cells [37]. 

We evaluated further the functional activity of astrocytes with dependency on the nanotopography that they interact with, by monitoring their spontaneous calcium activity on both continuous flat and nanostructured (Rq = 15 nm) zirconia. In these experiments, we found spontaneous calcium oscillations on several fields of confluent cells grown on continuous samples, where every astrocyte was in contact with many neighbouring cells. Calcium intensity traces (ΔF/F) acquired on cells cultured on flat and nanostructured homogenously covered samples as a function of time are shown in Figure 4; the corresponding time-lapse videos are reported in Appendix A. 

The mean integrated intensity per cell, calculated from the calcium intensity traces acquired during experiments run on different substrates, reveals a significantly larger value for cells on substrates with roughness 15 nm. The mean intensity per cell was 7.1 ± 3.7 (arb. units) for nanostructured ZrO_x_ with a roughness of 15 nm compared to a mean intensity compatible with zero, and 0.9 ± 1.2 (arb. units) on flat substrates. In general, cells on cluster-assembled substrates were found to show an enhanced spontaneous, single-cell calcium activity compared to cells on flat substrates, in agreement with results published in Ref. [70], where microgroove-patterned substrates (1 μm spacing grooves, with depths of 250 nm or 500 nm) were found to induce more frequent calcium transients in astrocytes. We confirm the same enhanced spontaneous calcium activity of astrocytes cultured on flat and nanostructured zirconia also at the physiological temperature (37 °C), as reported in Appendix A.

Altogether, these results demonstrate that the astrocyte/nanotopography interaction impacts the morphology, actin cytoskeleton organisation and functioning of these cells.

### 3.3. Pattern-Dependent Cell Network Properties

In their natural brain environment, astrocytes encounter and interact with elongated structures of micrometric dimensions, such as blood vessels and neuronal cells [55,56,57,58,59]. We therefore performed experiments with astrocyte cultures on micropatterned substrates with nanotopographical features that mimic such kinds of elongated geometries.

Figure 5a shows immunofluorescence images of the actin cytoskeletons of astrocytes confined on micropatterned substrates, a detail of astrocytes grown on a 50-μm-wide bridge (b), on a 50-μm-wide bridge (c) and on an isolated 100 μm dot (d).

From Figure 5, one can observe that astrocytes adhering to the bridges present an elongated morphology, which appear to be oriented parallel to the direction of the underlying bridge, as is the same with their actin components. By contrast, the cells on the dots (shown in Figure 5d) have no preferential orientation and they tend to occupy the entire area available. Astrocytes on the 50-μm-wide bridge are in an intermediate topographic configuration since they have enough room for spreading in all of the directions. Even astrocytic networks organize differently according to the region of the micrometric pattern that they grow on. On the bridges, the network is anisotropic; one cell statistically will find its first neighbours either on its left or right, toward the direction of the bridge, while the network has a circular shape on dots, where cells can make connections almost isotopically. 

We quantified the eccentricity of clusters of astrocytes in order to quantitatively describe this network organisation. In particular, the eccentricity of astrocyte clusters was calculated as the ratio between the width of the cluster along the direction of the bridges (w_x_) and the width of the cluster perpendicular to this direction (w_y_). Cells were selected in random locations on dots and bridges and for each cell a cluster of five cells was defined by locating the first four nearest neighbours. The location of each cell was identified as the centre of the nucleus, and w_x_ and w_y_ were taken as the distances between the two furthest cells in the direction parallel and perpendicular to the bridge. A schematic representation of the analysis procedure for the evaluation of clusters’ eccentricity is reported in Figure 6a,b. The average eccentricities of 15 clusters for each location (dot, 50 μm bridge and 20 μm bridge) selected from six different triplets of dots connected by bridges were calculated (the analysis was carried out on three independent samples). The results are reported in Table 1. 

Clusters located on dots are almost circular and they have no preferential orientation. Conversely, the eccentricity decreases for clusters on bridges of 50 μm and even more for those that are 20 μm large, thus quantifying the degree of cluster elongation toward the direction of the bridge. This is reasonable since astrocytes tend to occupy all of the adhesive surfaces available and bridges can only be occupied by a few cells close to each other in the axial direction of the bridge. There is in fact a chemico-physical constraint (the antifouling treatment) keeping the cells from occupying the entire surface of the cell culture substrate. It is thus possible to adjust the morphology and the actin cytoskeletal orientation of the single astrocytes and of their networks through a careful choice of the micrometric geometry of the pattern.

We quantified the density of cells on the different locations of the pattern via the analysis of immunofluorescence images of astrocytes in patterns. The spatial coordinates of the astrocytes were taken as the centre of the nuclei, stained with Hoechst 33342, and the locations of the cells on the different sections of patterns were manually discriminated (note that the cells at the border between dots and bridges were discarded from the analysis). A Kruskal–Wallis test [53] revealed a significant difference in the density of cells on dots, compared to the densities on 50 μm bridges and on 20 μm bridges. The boxplot in Figure 6c shows that the median density is lowest on dots, and highest on 20 μm bridges.

A multiple comparison using the Bonferroni method [54] revealed that the density of cells on dots is significantly different (*p*-value < 0.05) from the density on both bridges. On the contrary, the difference between the two bridges is not significant, although the median density increases as the bridge width decreases. 

This is an interesting result since astrocytes do not have a homogeneous population in the brain and they are characterised by different shapes according to their specific functionality [71,72]. As discussed in the introduction, we know that the shape of astrocytes and astrocytic networks is strongly affected by the morphological boundaries typical of the brain regions in which the cells develop. 

These experimental findings regarding the morphological changes induced by both the nanotopography and microconfinement suggest a future in-depth systematic study of GFAP and other astrocytic marker proteins (such as connexins 30 and 43; see below for the results with respect to the calcium activity) which can display a polarised expression and organisation in response to substrate stimuli.

To investigate the effect of the microscale confinement on Ca^2+^ activity in astrocytes, we performed Ca^2+^ imaging experiments on cells confined on zirconia nanostructured micropatterned substrates. We measured Fluo-4 fluorescent intensity integrated over time and the space of cells cultivated on different portions of the patterns, in particular, dots and 20-μm- and 50-μm-wide bridges. This method provides an estimate for the average concentration of intracellular calcium in the astrocytes, without considering the complex dynamics of the spontaneous calcium elevations. As discussed in the introduction, calcium activity in astrocytes can occur spontaneously, suggesting that these cells do not just provide responses to the neuronal synaptic activity but can drive it, acting as the primary source [73]. A higher integrated calcium intensity may be related to a higher predisposition of the cells to act as a source of astrocytic/neuronal activity.

We calculated the time-integrated calcium intensity for every cell located on the patterns, and also the average integrated intensity per cell over the different areas of the patterns, i.e., bridges and dots. A Kruskal–Wallis test [53] showed no significant statistical differences for the cells in the different locations and the integrated intensities per cell are comparable in the three topographic configurations. On average, the amount of intracellular calcium per cell is uniform all over the pattern. 

We may speculate that the integrated calcium intensity is not a good measure for evaluating the calcium activity of astrocytic networks, as it does not account for the dynamics of calcium influx. We therefore analysed a different parameter that includes the spatio-temporal features of calcium propagation: the calcium wave events. We defined a calcium wave event as the increase in the intracellular calcium concentration which is propagated to the neighbouring cells. A quantification of the number of calcium wave events occurring in cells located on bridges and dots on patterned substrates was carried out. The analysis was performed on the cell calcium traces extracted from the fluorescence videos. The details of the analysis are reported in Materials and Methods. In Figure 7, we report a graph with a series of traces, as an example, extracted from cells on dots connected by 20 μm bridges, where the different colours indicate different locations on the pattern; on the right, the representative snapshots acquired during time-lapse video are reported. The grey bars highlight the time intervals in which calcium waves take place, according to the calcium wave identification procedure described in the previous section.

Calcium elevations occurring on isolated traces are not considered to be calcium waves as they take place at the single cell. Furthermore, few cells display repetitive calcium elevations at frequencies of the order of tens of mHz, which can be classified as intracellular calcium oscillations, and they are also not comprised in the calcium wave classification. The density of calcium wave events per cell on dots of 50 μm and 20 μm bridges was defined as:(1)$$d_{e}=\overset{N_{e}}{\underset{i=0}{\sum }}\frac{n_{i}}{N}$$
where *N* is the total number of cells occupying the part of the pattern taken into consideration and *n_i_* is the number of cells involved in the i-th calcium wave event. This definition allows one to account for the number of calcium events and the number of cells participating in events; thus, we can say that a high *d_e_* corresponds to a more active region. The results of the analysis are reported in Figure 8.

A Kruskal–Wallis test [53] performed on such a number of calcium wave events revealed that there is a statistically significant difference between the density of events per cell in the different pattern locations. In particular, cells on 20 μm bridges conduct on average more calcium waves than cells on dots, with a significance level of *p* < 0.1 (calculated using a multiple comparison exploiting Bonferroni’s method [54]). In contrast, cells on bridges of different widths do not display significant differences in the dynamics of calcium elevations. This suggests that a difference in the calcium activity, and in particular the ability to propagate calcium waves, may be related to the astrocytic network level of gap-junctional coupling determined by different topologies. Interestingly, in homogeneous zirconia samples, every calcium signal recorded via time-lapse is uncorrelated, without wave propagation.

There is evidence that astrocytic networks are organised in a complex manner, with a controlled and variable number of connections, depending on the particular brain region [74]. In certain brain structures characterised by particular anatomic functional compartments, the shape of astrocytic circuits is dictated by functional units of neurons (such as in the somatosensory cortex [75] or the olfactory bulb [76]). Elsewhere, such as in the cortex or the hippocampus [28], it is the pattern of expression of connexins which determines the shape. So far, the functional relevance of these complex connectivity patterns and how they could impact calcium activity have not been clarified [77]. In this framework, computational approaches revealed a direct impact of network topology and calcium wave propagation, in the context of simulated astrocyte ensembles. Dokukina et al. [78] simulated a simple system of less than five cells in 2D, with the main communicating mechanism being IP3 diffusion through gap junctions. They showed that in these cell networks the variation in the spatial configuration of astrocytes determines different calcium responses. In particular, they identified as the main parameters the number of gap junctions per cell and the number of nearest neighbours. Consistent results were also found in 3D simulated astrocytic networks by Lallouette et al. [79], whose predictions hinted that calcium wave propagation is favoured in circuits that are sparsely coupled by gap junctions and when cell coupling at short distances is favoured. Our results agree with these computational models. In fact, the micropatterned substrates used in this work effectively impose limitations in the number of nearest neighbours a cell can have and promote cell coupling at short distances. In the future, studying the involvement of these gap-junction-dependent effects with dependency on the geometrical configuration of the astrocyte network will be interesting. 

The precise mechanism underlying spontaneous Ca^2+^ transients in astrocytes is still unclear [80]. One possible explanation is that Ca^2+^ events are triggered by stochastic Ca^2+^ entry through different pathways, such as Ca^2+^-permeable receptors, Ca^2+^ channels and Na^+^/Ca^2+^ exchangers [80]. When the intracellular Ca^2+^ concentration is high enough, Ca^2+^ release via IP3 receptors can be elicited, with the subsequent amplification and propagation of a calcium wave [26,81,82,83]. It has been demonstrated that stochastic Ca^2+^ oscillations in astrocytic processes are more likely when there is a high surface-to-volume ratio [84]. Hence, Ca^2+^ fluctuations are more likely to occur in the distal processes than in the cell soma. Cotrina et al. [85] demonstrated that an intact actin cytoskeleton is required for calcium oscillation and propagation. In fact, they showed that cytoskeletal organisation, and in particular the distribution of stress fibres, coincided temporally with the number of astrocytes engaged in calcium signalling after plating. 

Preliminary results (reported in Figure 9) of the elicited calcium activity of astrocyte, stimulated by a 50-ms-long stimulus of a 100 mM glutamate solution droplet (volume < 4 pL), applied 5 s after the beginning of the time-lapse video, suggest enhanced calcium activity on the bridges also in the case of elicited activity. Astrocytes are known to respond to glutamate, inducing the opening of ion channels and the subsequent rise in cytoplasmic free calcium, which can propagate to neighbouring cells [86].

We observed the propagation of calcium waves through neighbouring cells on the bridges, supporting the results obtained on spontaneous calcium activity. These data represent important preliminary results that indicate the potential of these micropatterned nanotopographical substrates for in-depth systematic characterisation of calcium signalling activity in microconfined astrocytic networks.

## 4. Conclusions

In this work, we have characterised the morphology, cytoskeletal architecture and the spontaneous calcium wave signalling of primary hippocampal astrocytes with dependency on nanotopographical substrate features and geometric confinement. We have shown that the interaction of astrocytes with nanostructured substrates favour changes in the cell shape towards peculiar morphological features and enhance the calcium activity. In addition, the geometric confinement of astrocytes via micropatterning these nanostructured surfaces was shown to further impact cell morphology and behaviour. Calcium wave signalling was only observed in astrocytes grown in these confined geometries, and was found to be more effective in cells forming elongated agglomerates with dimensions in the same range of blood vessels or axon fibres, i.e., geometrical features that astrocytes encounter in vivo. Our results are relevant as calcium oscillation and wave propagation have been proven to be closely related to astrocytic morphology [80] and actin cytoskeleton organisation [85,87,88,89]. The latter properties are themselves interdependent and are here shown to be impacted significantly by nanostructure. 

Our observations support the idea that there may be a relationship between Ca^2+^ dynamics and astrocyte morphology; the control of the network topography on the micro- and nanoscale offers the possibility to systematically characterize in vitro neuro-glial systems of increasing complexity, thus complementing animal models and brain slices, for studying the crosstalk between different brain cells in a pathological or physiological context. 

## Figures and Tables

**Figure 1 cells-12-00293-f001:**
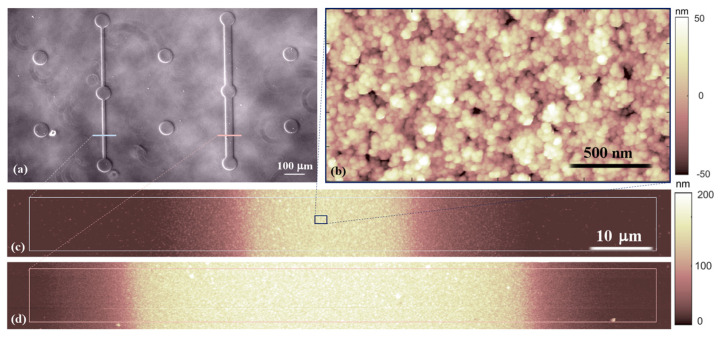
(**a**) Image of the micrometric nanostructured pattern of zirconia acquired using an optical microscope. Morphological maps of the nanostructured zirconia acquired via atomic force microscopy (AFM) in the homogeneous film (**b**), characterised by 15 nm surface roughness and sections of the bridges (20 and 50 µm large) connecting different islands (**c**,**d**).

**Figure 2 cells-12-00293-f002:**
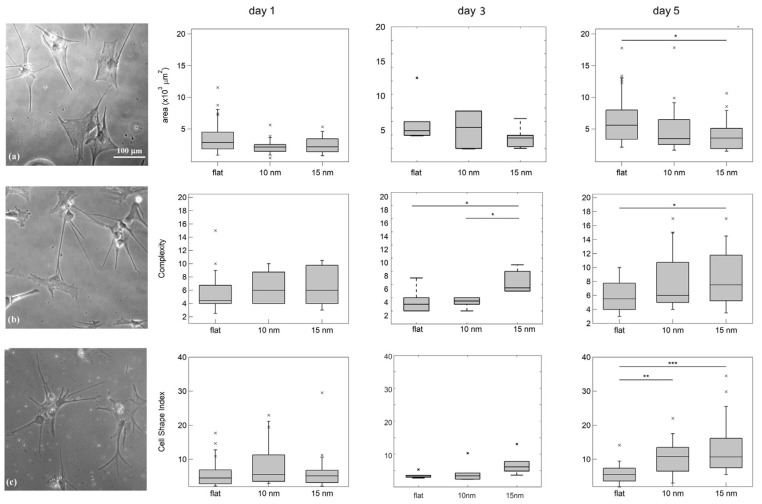
Optical images of astrocytes grown on flat zirconia (**a**), on 10-nm-rough zirconia (**b**) and on 15 nm of one (**c**) at 5 DIV. On the right are the morphological properties of astrocytes cultured on zirconia substrates with flat and nanorough (Rq = 10 nm or 15 nm) surface topography. The top boxplots display the cell area, the middle ones display the complexity parameter and the bottom ones display the CSI. The symbol * stands for outliers. All parameters are analysed after 1, 3 and 5 DIV.

**Figure 3 cells-12-00293-f003:**
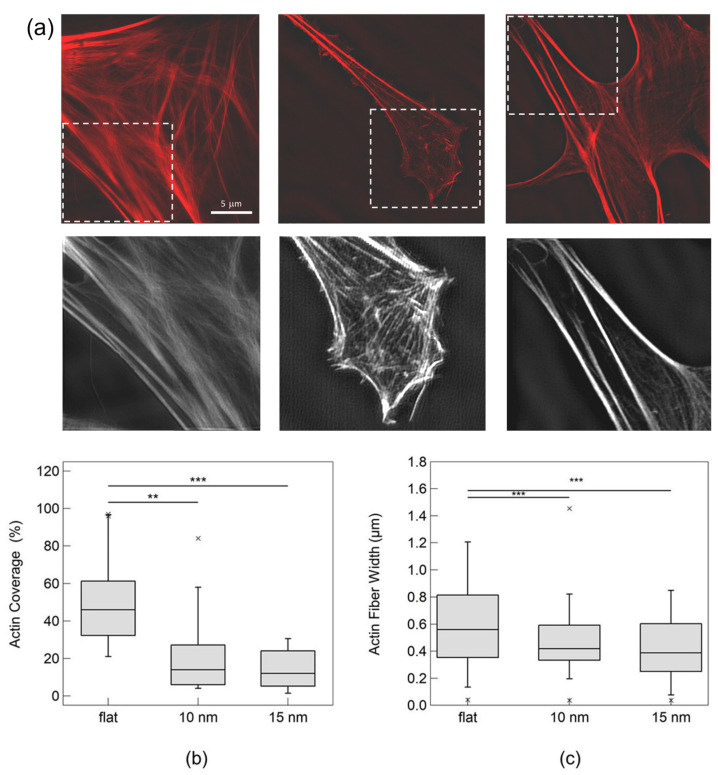
The actin cytoskeleton of astrocytes. (**a**) Immunofluorescence images of primary hippocampal astrocytes cultured on flat, 10-nm-rough and 15-nm-rough nanostructured zirconia substrates (from left to right), fixed and stained after 3 DIV. The actin cytoskeleton was stained red with phalloidin and tetramethylrhodamine (TRITC) conjugate. The symbol * stands for outliers. The images were taken using a 3D-SIM technique. Here, we show the maximum projection of the recorded stacks. Boxplot of the actin coverage (**b**) and actin fibre width (**c**) of astrocytes cultured on different zirconia substrates.

**Figure 4 cells-12-00293-f004:**
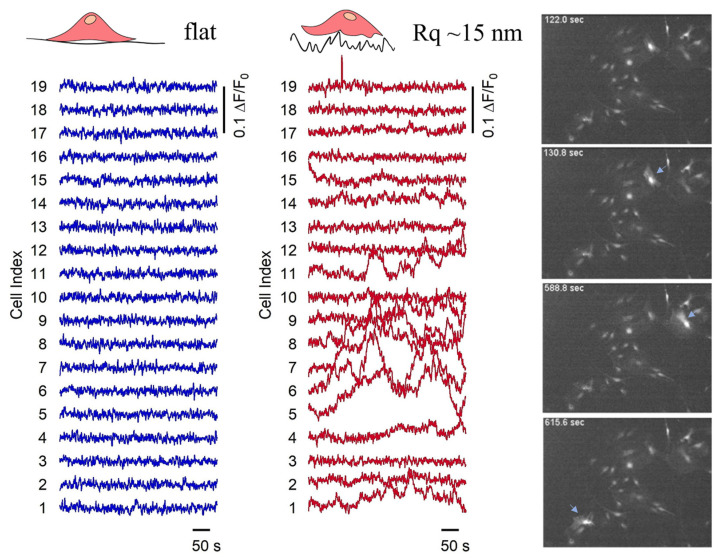
Calcium intensity traces (ΔF/F) of cells cultured on homogeneous zirconia samples, as a function of time, representing spontaneous activity of cells. Blue and red lines correspond to the traces acquired on flat and nanostructured (Rq~15 nm) zirconia thin film, accordingly; the videos they are extracted from are reported in Appendix A. On the right are representative snapshots of time-lapse videos recording spontaneous calcium activity of astrocytes cultured on 15-nm-rough zirconia substrate; light-blue arrows stress new calcium activity events.

**Figure 5 cells-12-00293-f005:**
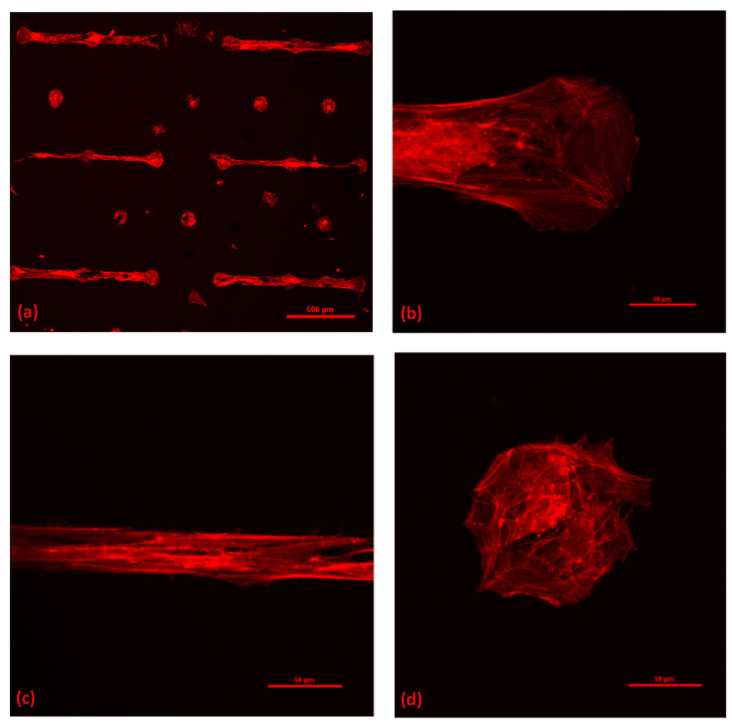
Fluorescence images of actin cytoskeletons of astrocytes confined on patterned nanostructured zirconia with bridges 20 and 50 µm large and dots 100 μm large in diameter (**a**), details of 50-µm-large bridge (**b**), 20-µm-large bridge (**c**) and 100 µm dot (**d**).

**Figure 6 cells-12-00293-f006:**
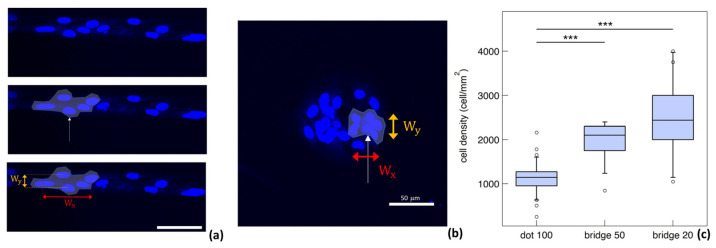
(**a**,**b**) Fluorescence images of nuclei of astrocytes confined on patterned nanostructured zirconia with bridges 20 µm large and dots 100 μm in diameter. In (**a**), we provide a schematic representation of the astrocyte cluster eccentricity analysis procedure: identification of a nucleus and its first 4 nearest neighbours, and calculation of the horizontal and vertical width of the cluster. In (**b**), the same procedure was performed for astrocytes cultured on nanostructured zirconia microdot. (**c**) Boxplot of the median density of astrocytes in different regions of the pattern. The symbol * stands for outliers.

**Figure 7 cells-12-00293-f007:**
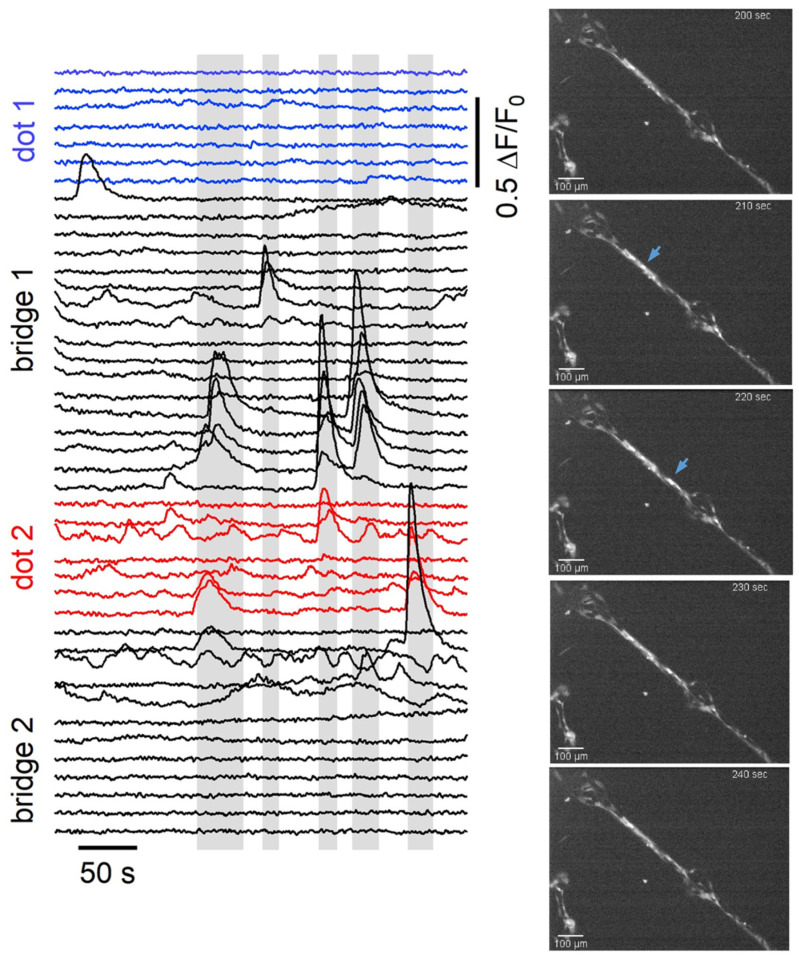
On the **left** are the calcium intensity traces (ΔF/F) of cells cultured on micropatterns as a function of time. Different colours indicate the different location of the cells corresponding to the traces. The grey bars highlight the calcium wave events, which can be recognised as the groups of intensity peaks on different cell traces. On the **right** are representative snapshots of a time-lapse video recording of spontaneous calcium activity of astrocytes cultured on micropatterned 15-nm-rough zirconia substrate. The video they are extracted from is reported in Appendix A.

**Figure 8 cells-12-00293-f008:**
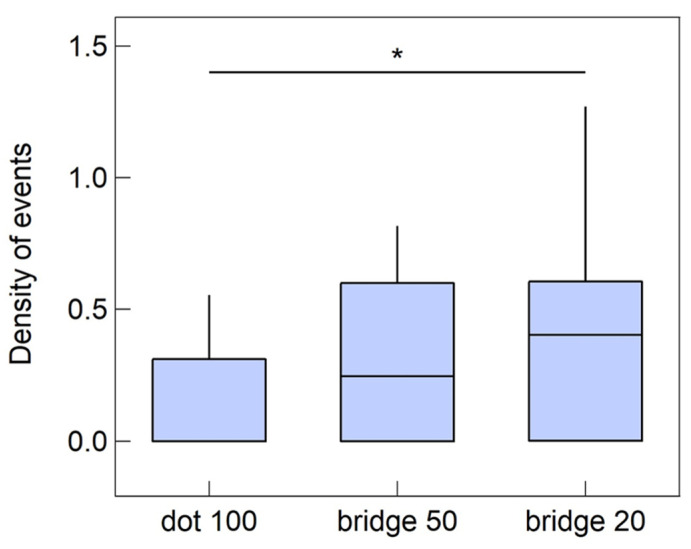
Density of calcium events as defined in the main text, for cells located on different areas of the patterned substrates, consisting of dots of 100 μm diameter connected by bridges with a width of either 50 μm or 20 μm. The symbol * stands for outliers.

**Figure 9 cells-12-00293-f009:**
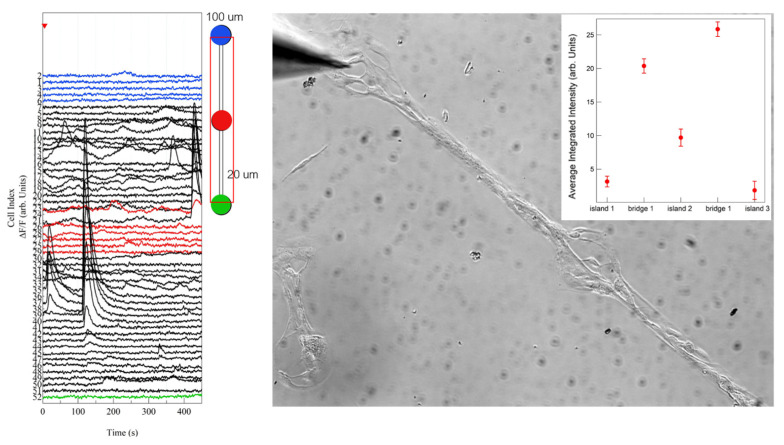
On the left are calcium intensity traces (ΔF/F) of cells cultured on micropatterns as a function of time after the addition of glutamate solution (after 5 s) close to the dot schematically represented by blue colour. Different colours indicate the different location of the cells corresponding to the traces ( blue, red and green traces correspond to cells on dots; black traces correspond to cells on bridges). In the inset, the averaged integrated intensity of calcium signals referring to the different zones is reported.

**Table 1 cells-12-00293-t001:** Eccentricity of astrocytic clusters grown on different locations of the micrometric patterns.

	Dot (100 µm Diameter)	50 μm Bridge	20 μm Bridge
Eccentricity	1.08 ± 0.23	0.80 ± 0.24	0.36 ± 0.16

## Data Availability

Data is contained within the article or Appendix A.

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
