# Peer review of "Nanotopography and Microconfinement Impact on Primary Hippocampal Astrocyte Morphology, Cytoskeleton and Spontaneous Calcium Wave Signalling"

_cells, 2023, doi:10.3390/cells12020293_

Round 1

Reviewer 1 Report

The authors have previously demonstrated the fabrication of complex micropatterns as tools to control cell adhesion and spatial distribution and used these patterns on various cell types, including primary astrocytes. In this work the authors investigated the impact of zirconia micropatterns on mechanotransduction and calcium signals in astrocytes. They found that growing astrocytes on nanostructured micropatterns modify their morphology, organization and calcium signaling. The work is interesting. However, some issues remain to be explored to strengthen the results.

Major points:

p. 5, l. 201: Which concentration do 2 ul of Fluo-4 AM correspond to?

p. 5, l. 203: Authors says that calcium imaging was performed at room temperature. Which is the reason behind this choice? Since cells are kept at 37°C in the incubator, calcium imaging experiments should be performed at the same temperature. Authors have to repeat the experiments at 37°C, which is a physiological temperature, to show the physiological relevance of their observations, especially because they insist that growing astrocytes on their zirconia micropatterns makes cultured astrocytes more similar to in vivo astrocytes.

p. 5, l. 212: The images of calcium signals were taken every 0.4 s. This is ok for slow signals. However, it has been shown that astrocytes can display faster calcium transients especially in their processes. As zirconia micropatterns increase astrocyte complexity, a higher frequency of image acquisition should be used in order to see whether moving from flat to nanostructured micropatterns also increases the richness of astrocyte calcium signals. This would increase the relevance of the observations.

p. 8, l. 333: The authors do not show data on day 3, but as they report that at day 3 there was an intermediate situation, it is important to show it as they did in figure 2, or at least as supplementary information

p. 8, l. 357. Why do the authors perform immunostaining for actin cytoskeleton at day 3, while the previous significant changes were observed at day 5 (see figure 3)? Authors should explain this choice or show data also at day 5, to be consistent with previous results.

p. 9, figure 4: from the provided images it is difficult to clearly see astrocytes with their cytoskeleton. Images with entire astrocytes need to be provided, together with a proper astrocyte labelling in addition to actin cytoskeleton (i.e. GFAP staining), to confirm that the cells observed and studied are astrocytes. Otherwise it is difficult to believe that the provided images are form astrocytes. The actual images can then be used as zoom to better show the actin cytoskeleton.

As nanostructured zirconia micropatterns modify the actin cytoskeleton of astrocytes, what about GFAP expression and organization? It is important to study this as the authors claim that micropatterns make astrocytes acquire a phenotype closer to in vivo one. Astrocytes on zirconia micropatterns display thicker actin fibers. It is interesting to study whether they increase GFAP expression as for reactive astrocytes in the brain.

p. 10, l. 402: As the authors refer to astrocyte properties on 50 um width bridges saying that there is an intermediate topographic configuration, they should show the data, either in the main figure or in supplementary materials.

p. 10, l. 412-416: The explanation on how eccentricity is calculated is difficult to follow and understand. Authors should provide a scheme to explain the method as well as representative images of astrocytes in the different conditions where we can appreciate the difference of eccentricity.

Beside the characterization of complexity and morphology and cytoskeleton of astrocytes, what is missing is the characterization of some classic astrocytic proteins which can display a polarized expression/localization, such as Cx43, AQP4 and Kir4.1. It is interesting to see whether the micropatterns can modify the expression and localization of these proteins in astrocytes, especially in correlation with the change in morphology. Do astrocytes grown on micropatterns express Cx30? Normally astrocytes in culture do not express Cx30 but it would be interesting to see whether micropatterns can modify this. Finally, what about astrocytes coupling via gap-junctions? Are astrocytes on micropatterns connected into larger networks? Dye coupling experiments will be useful to address this issue. This whole characterization will increase the relevance of the authors’ findings.

p. 13, figure 8: It is good to see traces of calcium signals, but an image of the astrocytic culture loaded with fluo-4 should be provided, together with a video (in supplementary material) of the acquisition.

The authors focused on spontaneous calcium events. What about triggered calcium signals (for example by exposing the cultures to ATP)?

Minor points:

p.1, l. 20: replace “cells network organization” by “network organization”.

p. 1, l. 38: replace “nerve cells” by “neurons”.

p. 1, l. 41: replace “neural” by “brain”.

p. 2, l. 47: replace “search for” by “sense”.

p. 2, l. 49: add “intercellular” before “networks.

p. 2, l. 53: replace “their” by “neuronal”.

p. 2, l. 54: replace “forms of signaling” by “signaling pathways”.

p. 2, l. 55: replace “changes” by “exchanges”.

p. 2, l. 57: replace “gap junction, which mediates” by “gap-junctions which mediate”. Add a sentence to explain which proteins (connexins) form gap-junctions in astrocytes (Cx43 and Cx30).

p. 2, l. 59: replace “generated” by “triggered”.

p.2, l. 78-84: the sentence is too long (6 lines) and it is not clear. Authors should simplify it, by dividing it into two shorter sentences.

p. 2, l. 85: replace “primary astrocyte confinement” by “the confinement of primary astrocytes”.

p.2, l. 88: replace “cells” by “astrocytes”.

p. 6, l. 275: 500 um or 500 nm? Please verify.

p. 7, l. 291: there is an error while referring to figure 2. Please correct.

p. 7, l. 315: replace “found in vitro cell cultures” with “found in in vitro cell cultures”.

p. 7, l. 317: replace “the lesion” with “brain lesion”.

p. 9, l. 358: there is an error while referring to figure 4. Please correct.

p. 9, l. 374: there is an error while referring to figure 4. Please correct.

Reviewer 2 Report

In this manuscript, Previdi et al., investigated the effects of the substrates with different micropatterns on in vitro cultured astrocytes. They utilized Zirconia nanostructured substrates to investigate the mechanotransdction events on astrocytes and they found nanostructure roughness has effects on astrocyte proliferation, cell morphology and cytoskeleton structures and cells network organization. They also discovered the nanostructure patterns affect the calcium waves. The manuscript showed the different effects of mechanical forces on astrocytes, however, some conclusions are not rigorous, there are some points should be addressed. Here are the details.

1.  The authors showed the different nanostructured substrates has effects on cell density of astrocytes, they claimed it is the effects on cell proliferation, however, it is not rigorous. It may affect the proliferation or and the cell death rate. The authors should use more direct manner like Edu or Brdu labeling to show the proliferation rate and the authors should show whether the nanostructured substrates have effects on cell death.

2. For figure 6, as the astrocytes showed dramatic different morphology and cell area, it doesn’t make sense to compare the cell density of astrocytes on dots and bridges. Could the authors compare the cells on bridges with different width?

3. For figure 7 and figure 8, could the authors provide the representative pictures to show the calcium influx in the cells?

Round 2

Reviewer 1 Report

The authors have addressed all my comments

Reviewer 2 Report

In the recent version of the manuscript, the authors have improved a lot. I am satisfied with the answers for the 1st revision as well.